# Detecting Defects in Composite Polymers by Using 3D Scanning Laser Doppler Vibrometry

**DOI:** 10.3390/ma15207176

**Published:** 2022-10-14

**Authors:** Daria A. Derusova, Vladimir P. Vavilov, Nikolay V. Druzhinin, Victor Y. Shpil’noi, Alexey N. Pestryakov

**Affiliations:** 1Industrial Tomography Center, Tomsk Polytechnic University, 634050 Tomsk, Russia; 2Research Lab. of Catalytic and Biomedical Technologies, Sevastopol State University, 299053 Sevastopol, Russia; 3Institute of Strength Physics and Materials Science, Siberian Branch, Russian Academy of Sciences, 634055 Tomsk, Russia

**Keywords:** nondestructive testing, 3D scanning laser Doppler vibrometry, local defect resonance, polymer composite

## Abstract

The technique of 3D scanning laser Doppler vibrometry has recently appeared as a promising tool of nondestructive evaluation of discontinuity-like defects in composite polymers. The use of the phenomenon of local defect resonance (LDR) allows intensifying vibrations in defect zones, which can reliably be detected by means of laser vibrometry. The resonance acoustic stimulation of structural defects in materials causes compression/tension deformations, which are essentially lower than the material tensile strength, thus proving a nondestructive character of the LDR technique. In this study, the propagation of elastic waves in composites and their interaction with structural inhomogeneities were analyzed by performing 3D scanning of vibrations in Fast Fourier Transform mode. At each scanning point, the in-plane (*x*, *y*) and out of plane (*z*) vibration components were analyzed. The acoustic stimulation was fulfilled by generating a frequency-modulated harmonic signal in the range from 50 Hz to 100 kHz. In the case of a reference plate with a flat bottom hole, the resonance frequencies for all (*x*, *y*, and *z*) components were identical. In the case of impact damage in a carbon fiber reinforced plastic sample, the predominant contribution into total vibrations was provided by compression/tension deformations (*x*, *y* vibration component) to compare with vibrations by the *z* coordinate. In general, inspection results were enhanced by analyzing total vibration patterns obtained by averaging results at some resonance frequencies.

## 1. Introduction

The last two decades have witnessed the appearance of several innovative polymer composites, the use of which has been broadening in industry, civil engineering, biomedicine, etc. [1,2]. Recent successes in tissue engineering and nanofiber composites have led to the elaboration of new medicines, biomedical scaffolds and implants. Nanofibers developed for environmental remediation purposes allow removing carcinogenic chemicals, acting in the form of filters for either oil spill control system or waste handling [2]. In general, the use of nanofibers [3,4], nanowires [5], nanotubes [6], films [7] and networks [8] have opened up new possibilities in the development of novel materials with various physico-chemical properties depending on particular application areas.

Novel construction composites uniquely combine high strength, low density and a good resistance against aggressive environment. A special emphasis is done on modification of natural and carbon fibers in composites used in mechanical engineering and aviation. Hybridization of synthetic and natural fibers ensures the enhancement of damping properties of natural fibers and cost reduction in manufacturing polymer glass- and carbon-fiber reinforced composites [9], even if they remain irreplaceable in some specific applications. Also, polymer composites are subjected to effective mechanical working being resistant against putrescence, fungal infection, mold, etc. [10].

However, polymer composites are relatively hygroscopic and susceptible to impact damage that is accompanied by the appearance of multiple in-depth cracks and delaminations. Therefore, the timely detection of manufacturing and in-service defects, as well as their quantitative characterization, represents a continuous challenge in the implementation of composites, particularly in high-tech industries [11,12].

Resonance laser vibrometry is a perspective field of ultrasonic nondestructive testing (NDT) of polymer composites due to its non-contact character and high sensitivity of vibration measurements. The combination of acoustic stimulation with laser scanning allows for the visualizing of the propagation of elastic waves in materials, including their interaction with structural defects [11,12]. By using the technique of scanning laser Doppler vibrometry (SLDV), it was shown elsewhere that internal defects distort the front of propagating acoustic waves, thus providing specific defect signatures [13,14,15]. Moreover, structural inhomogeneities enhance vibrations in defects subjected to ultrasonic stimulation that can be explained in the framework of the Local Defect Resonance (LDR) concept. The respective inspection technique involves the optimization of ultrasonic wave frequencies by making them equal or close to the mechanical resonance frequencies of a defect, thus substantially enhancing vibration amplitudes in the defect. This phenomenon is caused by a local decrease in the rigidity of parts, which respond to ultrasonic excitation [16]. In fact, the resonance pumping of acoustic energy into materials under test improves the efficiency of defect detection when using individual laser stimulation and laser vibrometry. Many aspects of using LDR for detecting defects of various origins were reported in [13,14,15].

Basic investigations of resonance phenomena in objects with structural defects subjected to acoustic stimulation were conducted on polymer materials. To facilitate analytical calculations, samples of simple geometric forms were analyzed by using the classical oscillation theory [17]. The typical surrogates of structural defects are flat bottom holes (FBH) often made in polymethylmetacrylate (PMMA) samples. In this case, a partial lack of the material in defect zones models a local deterioration of the sample rigidity, and the material layers over defects are considered for calculating resonance frequencies. Analytical expressions for determining resonance frequencies were obtained for circular and rectangular defects [16]. The fundamental LDR frequencies were determined for planar defects of various shapes in polymer/composite materials by using the classical Rayleigh method [15]. The derived formulas indicate that LDR frequencies depend on defect size and shape, as well as material properties, and acoustic waves of particular frequencies can be used for identifying defects of different shapes. The suggested analytical expressions were experimentally verified to reveal a good match (with errors under some percent) in determining basic resonance frequencies.

It is important that the appearance of resonance phenomena in defects is accompanied by local heat generation and respective temperature elevations which are of a vibrational nature. The combination of in-plane (*x*, *y*) LDR with vibrothermography was used for the inspection of barely visible impact damage in carbon fiber reinforced polymers (CFRP) [18,19]. It was shown that excitation of in-plane LDR frequencies results in increased thermal contrasts in defect zones compared to the out-of-plane (*z*) LDR frequencies. This allows using excitation ultrasonic sources of a lower power. In general, the phenomenon of LDR ensures a significant reduction of applied ultrasonic power in regard to the so-called ultrasonic infrared (IR) thermography requiring stimulation sources with electric power up to a few kW [20,21,22].

The contribution of *x*, *y*, *z* vibration components into a total LDR vibration pattern was analyzed in some studies, in particular, in application to impact damage detection [19,20]. However, in the case of real defects, vibration patterns are fairly complex to perform a decent comparison of the contribution of individual components. In this investigation, three components of vibration components are comparatively analyzed on defects with a simple geometry by modeling local resonance vibrations and performing 3D laser scanning. Such an approach has allowed for the evaluating of stresses introduced in the samples as a result of resonance ultrasonic stimulation in comparison with the material yield strength. In order to detect, localize, and visualize complex defects in multi-layer composite materials by using the technique of 3D SLDV, the experimental research included detection of impact damage in a cross-ply CFRP composite.

## 2. Materials and Methods

A rectangular FBH (32.5 × 31 × 0.9 mm) located in a PMMA reference sample (200 × 200 × 4 mm) has been used as a defect surrogate to study three components of vibrations appearing as a result of resonance ultrasonic stimulation (see the sample geometry in Figure 1). It is worth mentioning that sample milling quality, as well as material roughness and weak variations in defect geometry, may essentially influence experimental results. Mechanical vibrations in the sample were analyzed by applying the technique of 3D scanning laser vibrometry in combination with wide-frequency range ultrasonic stimulation. The experimental setup included a scanning laser vibrometer PSV-500-3D-HV (Polytec, Waldbronn, Germany) and a signal generator AWG-4163 (Aktakom, Novosibirsk, Russia), which supplied acoustic signals via an amplifier AVA-1810 (Aktakom) to a piezoelectric transducer attached to the sample (Figure 2a).

SLDV is typically used for the non-contact evaluation of vibrating object velocity being characterized by a high data accuracy and precision. The vibration components in the in-plane and out-of-plane directions are experimentally determined at each scan point. However, it is worth noting that 3D vibration measurements should be carried out at small angles [23] to improve the accuracy of vibration evaluation (up to nm) and reduce noise phenomena. In some applications, it is advisable to use an infrared laser-based vibrometer allowing to take material reflectivity into account at large distances (up to 125 m) [19].

In this study, the propagation of elastic waves in the reference sample and their interaction with structural inhomogeneities were analyzed by performing 3D scanning of vibrations in the Fast Fourier Transform (FFT) mode. The scanning step was 31.25 Hz, thus resulting in 3200 spectral lines in total. The vibration velocity on the surface of the sample were measured with a scanning step of 2 mm. At each scanning point, the in-plane (*x*, *y*) and out of plane (*z*) vibration components were analyzed. The acoustic stimulation was fulfilled by generating a frequency-modulated harmonic signal in the range of 50 Hz to 100 kHz. In addition, the results of repeating measurements were averaged at each scan point prior moving the laser beam to the next point. Note that the good repeatability of non-contact measurements was confirmed by using the laser vibrometer characterized by a high stability of measurement parameters in time.

## 3. Modeling

The development of a proper FBH model simulating vibrational characteristics of real objects is useful in order to minimize the size of experiments and correct some object properties, such as mass/size ratio and rigidity, and/or damping characteristics that may be necessary for altering object resonance frequencies. The analysis of periodic vibration parameters (frequency and shape) appearing in an elastic medium is useful for evaluating the resistance of tested parts against external mechanical impacts and, in some cases, for performing the correction of part work frequencies.

The FEM modeling of vibrations in the PMMA plate was based on the Eigenfrequency Analysis by using the Structural Mechanics Comsol Multiphysics package. The model included Solid-type volume elements with the properties of the tested material, see [24] for details. In general, the calculation of plate natural frequencies requires the knowledge of plate bending stiffness (*D*) [24]:(1)$$D=\frac{Eh^{3}}{12\left(1-v^{2}\right)}$$
where *h* is the FBH thickness, *E* and *v* are the Young’s modulus and Poisson ratio of material, correspondingly.

The eigenfrequencies, as well as the eigenmodes, depend on the sample geometry and on support conditions on the edges. In the case of a rectangular FBH with the simple support and side lengths *a* × *b*, the eigenfrequencies can be found as [24]:(2)$$\omega_{m,n}=\pi^{2}\left(\frac{m^{2}}{a^{2}}+\frac{n^{2}}{b^{2}}\right)\sqrt{\frac{D}{µ}},$$
where *µ* is the mass per unit area, *m* and *n* are the indices (1, 2, 3, etc.) denoting the resonance mode numbers.

The analytical equation for determining the fundamental LDR frequency of a rectangular FBH in an isotropic PMMA plate was recently suggested in the form [17]:(3)$$\omega_{0}=\frac{2\pi^{2}}{3a^{2}b^{2}}\sqrt{\frac{Eh^{2}\left(3a^{4}+3b^{4}+2a^{2}b^{2}\right)}{3\rho \left(1-\nu^{2}\right)}},$$
where ρ is the material density.

The eigenfrequencies of a rectangular FBH in a PMMA plate were calculated by Equations (1)–(3) for the following experimental parameters: *a* = 32.5 mm, *b* = 31 mm, *h* = 0.9 mm. The material mechanical properties were: *E* = 3 × 10^9^ Pa, *v* = 0.4 and ρ = 1160 kg/m^3^. Figure 3 shows the used finite-element model (Figure 3a) and the respective numerical mesh (Figure 3b) with the square FBH being the area of interest; the computation time was 10 s.

The results of calculation were values of FBH resonance frequencies *f_model_* and distributions of mechanical vibrations by *x*, *y*, *z* coordinates. In the model, the boundary condition involved the cantilever support, thus accurately reflecting the real experiment. Also, the Eigenfrequency Analysis assumed that the model was dynamically tested in ideal conditions, i.e., without taking into account possible external factors. The experimental values of the resonance frequency *f_exp_* determined by means of laser vibrometry were used to validate the mathematical model.

## 4. Results and Discussion

### 4.1. FBH Tests

The results of performing laser vibrometry in the FFT mode were the amplitude-frequency spectra of the PMMA plate (Figure 4) and the images of material surface vibrations in the range from 50 Hz to 11 kHz obtained with the scanning step of 31.25 Hz.

At each resonance frequency, the magnitude of vibration amplification was evaluated by determining the signal-to-noise ratio (*SNR*) that is a common practice in comparing the efficiency of various experimental and data processing techniques. The *SNR* approach evaluates signals in contrast to background and/or system noise [25]:$$SNR=\frac{\left|\bar{v_{d}}-\bar{v_{nd}}\right|}{\sigma_{nd}},$$
where $v_{d}$ and $v_{nd}$ are the amplitudes of vibration velocity in defect and non-defect areas, and *σ_nd_* is the standard deviation of $v_{nd\;}$.

It is seen from Figure 4 that the frequency spectra by *x*, *y*, *z* coordinates look similar by shape but different by amplitude. For each coordinate, the mean vibration amplitudes were calculated in chosen frequency bands and then compared to exhibit the contribution of each coordinate into the resulting vibration picture. The mean vibration velocity is maximal by the *z* coordinate being 300 mm·s^−1^ to compare to 130 and 230 mm·s^−1^ by the *x* and *y* coordinates respectively. This demonstrates the maximal contribution of bending deformation in the case of the resonance stimulation of FBH-like defects. It is also worth noting that the resonance frequencies for all three components are identical. In other words, the amplification of vibrations occurred by the *x*, *y*, *z* coordinates at the same frequencies, or frequency bands, and following the same order of resonance frequencies. For example, the *x*, *y*, *z* vibrations of the fundamental resonance in the particular FBH appeared in the frequency range from 2344 to 2750 Hz (Figure 5). In this range, the maximal amplification of vibrations (*SNR* = 35) took place by the *z* coordinate (out-of-plane LDR) while, by the *x* and *y* coordinates (in-plane LDR), the maximal *SNR* values were only 12 and 24, respectively.

In spite of the fact that LDR frequencies differ by three coordinates, all three vibration components are equally involved in the process of vibrations. This is also confirmed by the identical distribution of vibration velocities along three coordinates in the amplitude-frequency characteristic (AFC) of the PMMA plate (Figure 4). By having analyzed the AFC of the plate in the frequency range from 50 Hz to 13 kHz, three resonance frequencies were identified over the rectangular FBH by each Cartesian coordinate. The shapes of resonance vibrations were considered separately and compared at each particular LDR frequency. The results of the analysis are presented in Table 1.

The SLDV results have demonstrated that the maximal amplification of wave front vibrations from 0.07 to 0.6 mm·s^−1^, i.e., by nine times, appeared at the defect fundamental resonance frequency (2750 Hz) by the *z* coordinate (out-of-plane LDR). The vibrations by the *x* and *y* coordinates (in-plane LDR) have been amplified only by three and four times, respectively. This confirms the conclusion that out-of-plane vibration components play a decisive role in the build-up of resulting defect signals at fundamental LDR frequencies. For example, when considering the FBH resonance of a higher-order, it was found that the contribution of single vibration components into the total vibration pattern was non-uniform. The maximal vibration velocity by the *x* coordinate (up to 1.4 mm·s^−1^) was observed at the frequency of 6094 Hz being higher than vibrations by the *z* (1.1 mm·s^−1^) and *y* (0.9 mm·s^−1^) coordinates. However, all three components contributed to the total vibration pattern obtained by averaging the components at the frequency of 6094 Hz. It is believed that such an LDR feature may be useful when performing the NDT of materials and parts. For instance, in the test case considered above, the summation of *x*, *y*, and *z* vibrations at particular frequencies made the corresponding defect indication closer to the real defect by size and shape. Furthermore, by analyzing vibrations at the third resonance frequency of 9250 Hz, the maximal contribution was provided by the *x*-component (in-plane LDR) to compare to the *y*- and *z*-components. Therefore, the *x*, *y*, *z* vibration pattern at 9250 Hz resembled the pattern obtained by *x* and *y* coordinates, i.e., when the contribution of out-of-plane vibrations was minimal. In general, it seems that, even if individual vibration components can be different at each particular frequency, the magnitude of total vibrations remains comparable to both in-plane and out-of-plane LDR components. The total vibration patterns obtained for each spatial coordinate showed the defect indication which is square-shaped and close in size to the true FBH. It is believed that total vibration patterns obtained by three coordinates in the mode of 3D scanning may supply the most reliable information about FBH parameters, including maximal *SNR* values, thus enhancing the efficiency of defect detection.

The results of numerical modeling illustrate that calculated FBH vibration shapes averaged by three coordinates repeat those by the *z* coordinate. This is explained by the predominant contribution of *z*-vibration amplitudes into the total pattern. Therefore, the comparison of theoretical and experimental results on determining vibration shapes and resonance frequencies has been performed for the *z*-component.

As follows from Table 1, the LDR vibrations at each frequency appeared by all three coordinates being slightly different by *x* and *y* coordinates. This phenomenon was described elsewhere to illustrate that polarization of different LDR mode shapes with respect to defect orientation may have a considerable effect on defect detectability [18]. The in-plane LDR takes place due to a vibration mode produced by waves of a higher velocity. It is worth mentioning that frictional heating plays a predominant role in the signal buildup when performing ultrasonic vibrothermography.

By taking into account the above-mentioned phenomena, the first three modes of FBH resonance vibrations were identified in the frequency range from 2750 to 9500 Hz to reveal a good agreement with the theoretical data by shapes and frequencies of resonance harmonics. The first calculated mode at the frequency of 2690 Hz corresponded to the experimental results of the scanning at the frequency of 2750 Hz (the second mode of the LDR frequency was characterized by the theoretical/experimental values of 5030 and 6094 Hz, respectively). It is worth noting that calculated shapes of resonance vibrations in total patterns were different in regard to the experimental *x*, *y* component values. In fact, the calculated total pattern repeated vibrations by the *z* coordinate while the experimental pattern combined vibrations by the *x* and *z* coordinates. A certain discrepancy between simulated and experimental results (up to 10% by the central frequency) was also observed at the higher frequencies. It is believed that such a discrepancy appeared because of different conditions of inputting ultrasonic waves into the sample and some peculiarities of FBH manufacturing, namely, the presence of rounded corners, high material roughness, etc. Moreover, the laser scanning was performed in the case, where the finite-size piezoelectric transducer was attached to the sample edge (Figure 2b), but the modeling assumed the ideal condition without taking into account any environmental factors. However, in spite of the above-mentioned discrepancies between the theory and experiment, the obtained results matched reasonably well to confirm the validity of the proposed mathematical model.

### 4.2. Analyzing Acoustic Stimulation Power

As mentioned above, the resonance acoustic stimulation of defective composite samples causes the essential amplification of vibrations in defect zones. The sample material experiences maximal compression/tensile displacements in these zones. In this study, the maximal stresses appearing in the FBH under resonance acoustic stimulation were evaluated. The PMMA plate (Figure 1) was stimulated at 1250 Hz, which was the fundamental LDR frequency for the rectangular FBH in this sample. The surface in-plane vibrational response by the *x* coordinate was investigated using a 3D scanning laser Doppler vibrometer (Figure 2), and the magnitude of applied stresses accompanying acoustic resonance stimulation was compared with the PMMA tensile strength.

As a result of laser scanning, the absolute sample displacement of 0.7–1 m occurring during compression/tension of FBHs (length *L*) was determined by the *x* coordinate. This value corresponded to the in-plane component of vibrations (normal strain).

Suppose that the stresses, which appear under ultrasonic stimulation, are essentially lower than the corresponding strength limit. In this case, all deformations will appear in the elastic regions, and the magnitude of the stresses accompanying acoustic resonance stimulation should be compared with the PMMA tensile strength. By introducing the relative displacement *ε* (*Δl*/*L*) and the Young modulus *E*, the normal stress *σ* can be found as *σ* = *E* × *ε*. Next, one can evaluate vibrations and the relative displacement by *x* as follows: *Δl*/*L* = (0.7 ÷ 1) m/31 mm = (0.023 ÷ 0.032) × 10^−3^. For PMMA, the Young’s modulus is 2800–3300 MPa [23], and the maximal stress appearing in the FBH under ultrasonic stimulation is: *σ_exp_* = (2800 ÷ 3300) MPa × (0.023 ÷ 0.032) × 10^−3^ = (64.4 ÷ 105.6) kPa. Furthermore, the PMMA tensile strength is *σ_max_* = 65 MPa [26]. The ratio between the tensile strength of PMMA material and maximal stresses appearing in the FBH under ultrasonic stimulation is *σ_max_*/*σ_exp_* = 65 MPa/(64.4 ÷ 105.6) kPa = 615 ÷ 1000. The estimates obtained confirm a nondestructive character of ultrasonic resonance stimulation of such composite materials, which are close to PMMA by their mechanical properties.

### 4.3. 3D Scanning of Impact Damage in CFRP

It was shown elsewhere that vibration patterns allow the visualization of defect contours and, hence, the evaluation of defect lateral size and shape by applying acoustic excitation at the corresponding resonance frequencies [26], as well as using sweep excitation in the low- and mid-kilohertz range [27,28,29]. Defect identification can be enhanced by analyzing a total vibration pattern obtained by averaging results at some resonance frequencies. Following this concept, a 1.5 mm-thick CFRP plate containing an 18 J impact damage was inspected (Figure 6), and the laser vibrometry results were validated by the results of the ultrasonic C-scan test. An ultrasonic inspection was performed by using an Ideal System 3D scanner supplied with a focused transducer (resonance frequency 15 MHz, diameter 16 mm, focusing depth 50.8 mm and scanning step 1 mm). The results of ultrasonic inspection were presented as C-scans (see Figure 7).

The results of ultrasonic C-scanning have revealed the known structure of impact damage in a cross-ply CFRP composite including multiple cracks and delaminations of which area increases with depth counted from the front surface. Often one considers the cone-, or tree-, or pyramid-like shapes of impact damage. The minimal delamination with the area of 175 mm^2^ was identified at the depth of 0.1 mm, i.e., directly under the point of impact (Figure 7a), while the largest delamination (area of 1530 mm^2^) was detected at the depth of 1.4 mm (Figure 7b). The *SNR* values were 7.6 and 14 in these two cases, respectively.

As mentioned above, the efficiency of laser vibrometry can be enhanced by determining defect resonance frequencies. Performing laser vibrometry in the FFT mode allows for the transforming of signals from the time to phase/frequency domain, thus enabling, first, the analysis of vibrations at individual frequencies and, second, the averaging of the results across the total spectrum. The use of such an approach for performing the spectral analysis of impact damage was described elsewhere [16]. In our study, a 1.5 mm-thick CFRP plate was investigated by using laser vibrometry in combination with a contact transducer mounted on the sample surface, as shown in Figure 6. Up to 20 frequencies were identified in the range from 185 Hz to 100 kHz, where various defect sections revealed particular resonances. The maximal mean amplitude of vibration velocity (0.34 mm·s^−1^) was observed by the *y* coordinate being slightly higher than that by the *x* and *z* coordinates (0.23 mm·s^−1^ in both cases). This demonstrates the domination of compression/tension deformations in the case of resonance stimulation, even if vibration amplitudes are close by all three coordinates. Figure 8 illustrates the amplification of vibrations in the impact damage zone at some defect resonance frequencies (7.44 kHz, 32.8 kHz and 46.5 kHz), thus proving that resonance vibrations appear in a wide range of frequencies.

Another important factor influencing a picture of vibrations in defects is the contribution of each vibration component into the total signal. Similar to the FBH case, the appearance of resonance at some particular frequencies leads to the amplification of vibrations by *x*, *y*, *z* coordinates. However, different sections of impact damage are characterized by different shapes and amplitudes of vibrations. This statement is illustrated in Figure 9, where contributions of in-plane and out-of-plane vibration components into the total pattern are shown for the particular LDR frequency (43.6 kHz).

The images presented in Figure 9 only provide an example of the contribution of the *x*, *y*, *z* components into the resulting signal at a particular resonance frequency, but they allow for the understanding of the basics of resonance phenomena in materials with structural defects. It is seen that, at particular LDR frequencies, the enhancement of vibration amplitudes in defect zones takes place by all Cartesian coordinates. Material deformation occurs by both out-of-plane and in-plane coordinates being governed by the same physical phenomena. Therefore, it can be assumed that LDR frequencies correspond to defect vibrations, which simultaneously appear by all three coordinates and should be considered as a single process.

Figure 10 shows the images of CFRP surface vibrations averaged across the total frequency spectrum for three vibration components. The defect area determined on the sample rear surface by means of 3D laser vibrometry was 1760 mm^2^, or 85% in regard to that measured ultrasonically (compare Figure 7b and Figure 10). It appears that supplying a frequency-modulated causes vibrations of a greater defect area by activating vibrations at both the basic and higher defect resonance frequencies. Therefore, the determination of defect size and shape will be more accurate by applying a wide-range acoustic stimulation. Figure 10 illustrates that the amplitude of vibration velocity by the *x*, *y* coordinates reaches 50 mm·s^−1^, while by the *z* coordinate it only reaches 15 mm·s^−1^. Furthermore, in-plane (*x*, *y*) components of vibration ensure a greater area of vibrations to compare to the out-of-plane (*z*) component. Hence, unlike the case of FBHs with maximal out-of-plane resonance vibrations, in CFRP, the predominant contribution to the averaged vibration pattern of impact damage is provided by deformations of compression/tension, i.e., in-plane components of vibrations. In the case of FBHs, this can be explained by higher vibrations along the coordinate *z*, which accompany the removal of some mass of material. In impact damage defects, the predominance of *x*, *y* vibrations leads to the prevailing tangential interaction of in-plane interfaces and respectively to local heat generation in crack-like defects [18].

## 5. Conclusions

Peculiarities of vibration patterns in two reference samples containing a simple FBH-like defect and an 18 J impact damage of a complex shape have been investigated by applying the technique of 3D scanning Doppler laser vibrometry. In the PMMA sample containing a rectangular FBH, the resonance vibrations appeared by three Cartesian coordinates (*x*, *y*, *z*) at each resonance frequency, but their contribution varied at different harmonics. For instance, at the frequency of 2.75 kHz, the first mode of resonance vibrations appeared by the *z* coordinate, while the second mode was significant by the *x*, *y* coordinates. Such peculiarity of resonance vibrations in the FBH-like defect was observed at all three resonance frequencies in the range from 2.75 to 9.25 kHz. It is believed that the summing of vibrations by three coordinates allows a better identification of defect parameters compared to the vibration analysis by one coordinate only.

The analysis of total vibration patterns by each coordinate showed that the surface distribution of the corresponding frequency data was of a rectangular shape, thus being close to the true FBH shape. However, the corresponding three patterns vary in shape. It has been suggested that the most reliable information about defect size and shape is supplied by the total vibration pattern obtained by summing the frequency band data along three coordinates across the spectrum of interest. In addition, such patterns ensure the maximal signal-to-noise ratio when measuring vibration velocity on the noisy background.

The finite-element analysis of resonance phenomena occurring in FBHs under acoustic stimulation has allowed for the minimizing of the size of the experiments by comparing numerical and laser vibrometry results across the spectrum of resonance frequencies, as well as the analyzing of frequency distribution along three spatial coordinates. The discrepancy between the theoretical and experimental data has not exceeded 10%.

By assessing the nondestructive character of the LDR technique, it has been found that the ratio between the tensile strength of the material tested and the amplitude of applied ultrasonic stimulation is from 615 to 1000.

Under resonance stimulation, the FBH-like defects are characterized by the prevailing contribution of bending deformations into the total vibration patterns, i.e., by the out-of plane (*z*) vibration component. However, in the case of impact damage in CFRP, it has been shown that the predominant contribution into total vibrations is provided by compression/tension deformations (*x*, *y* vibration component) to compare with vibrations by the *z* coordinate. The apparent defect area determined by the in-plane (*x*, *y*) vibrations has been closer to the true one than the estimate obtained by the out-of-plane (*z*) component.

To summarize, the efficiency of laser vibrometry can be enhanced by evaluating defect resonance frequencies. Performing laser vibrometry in the FFT mode allows for the transforming of signals from the time to phase/frequency domain, thus enabling the analysis of vibrations at individual frequencies and then the averaging of the results across the total spectrum.

Future research will be devoted to the analysis of resonance frequencies appearing in defects of a complex shape, such as single delaminations and impact damage in glass and carbon fiber reinforced composites. The relationship between the frequency spectrum composition and impact damage energy is to be investigated to better understand the potentials of SLDV in practical evaluation of damage severity, particularly in aviation applications. In addition, a deeper theoretical model will include 3D scanning.

## Figures and Tables

**Figure 1 materials-15-07176-f001:**
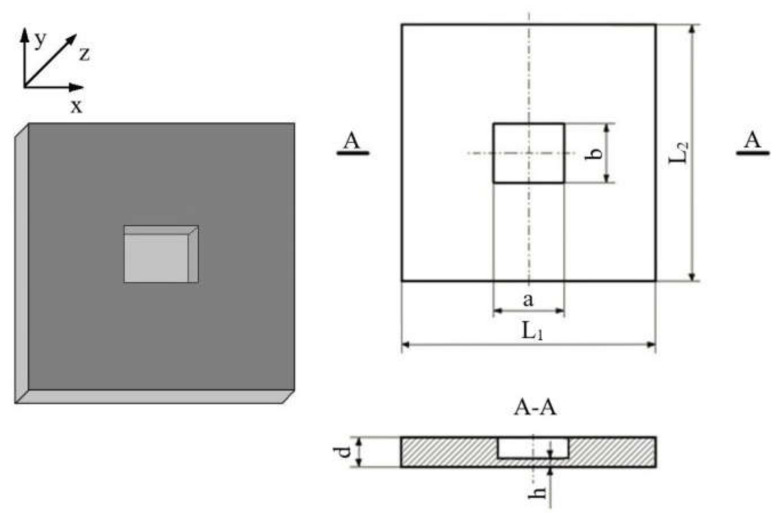
General view and geometry of PMMA sample weakened by rectangular FBH.

**Figure 2 materials-15-07176-f002:**
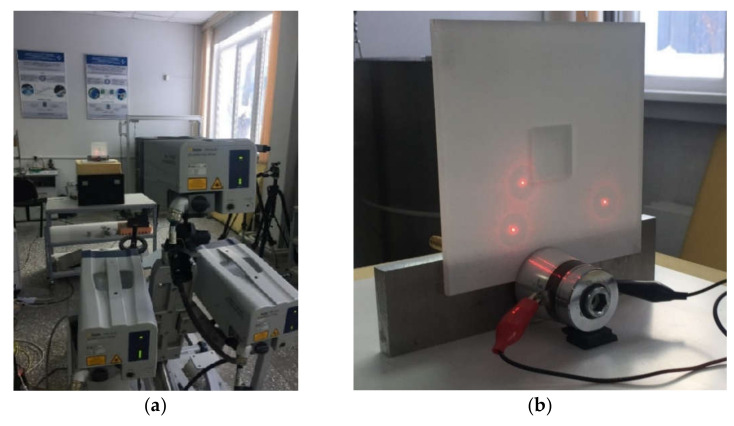
3D scanning laser Doppler vibrometer experimental set-up (**a**) and piezoelectric transducer attached to sample (**b**).

**Figure 3 materials-15-07176-f003:**
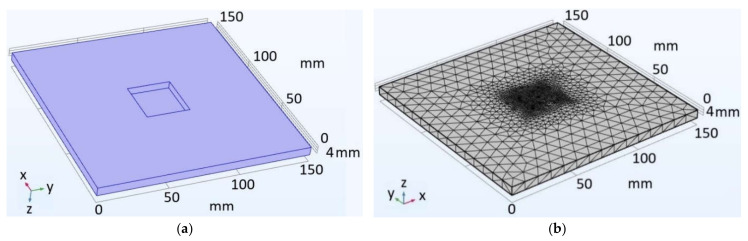
Finite-element FBH model (**a**) and numerical mesh (**b**).

**Figure 4 materials-15-07176-f004:**
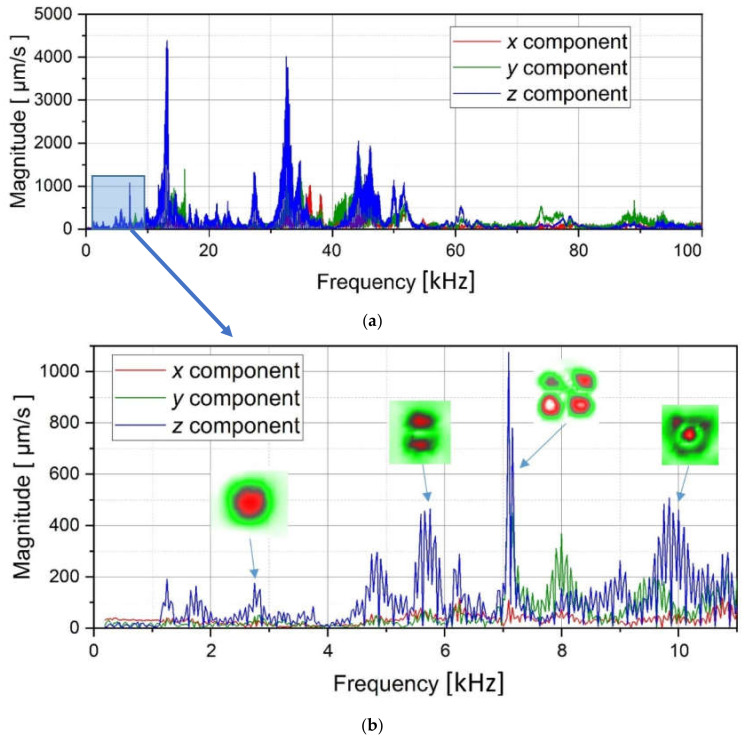
Total amplitude-frequency spectra (**a**) and their magnified section (**b**) for PMMA plate activated by acoustic signals in the range from 50 Hz to 11 kHz.

**Figure 5 materials-15-07176-f005:**
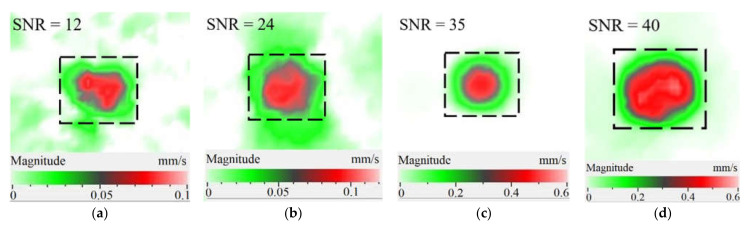
LDR vibrations over rectangular FBH: (**a**)—by *x* coordinate (2344 Hz frequency), (**b**)—*y* (2563 Hz), (**c**)—*z* (2750 Hz), (**d**)—frequency band data (2334–2750 Hz).

**Figure 6 materials-15-07176-f006:**
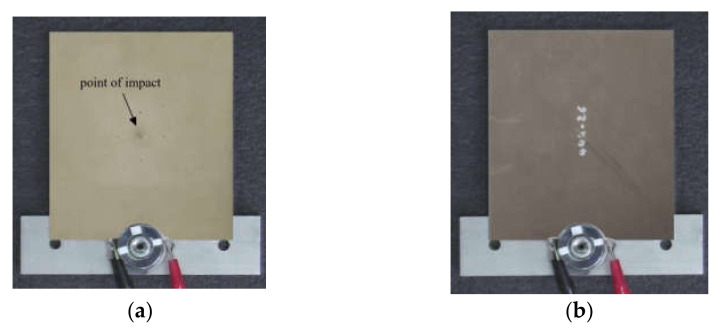
CFRP sample with 18 J impact damage: (**a**)-front surface, (**b**)-rear surface.

**Figure 7 materials-15-07176-f007:**
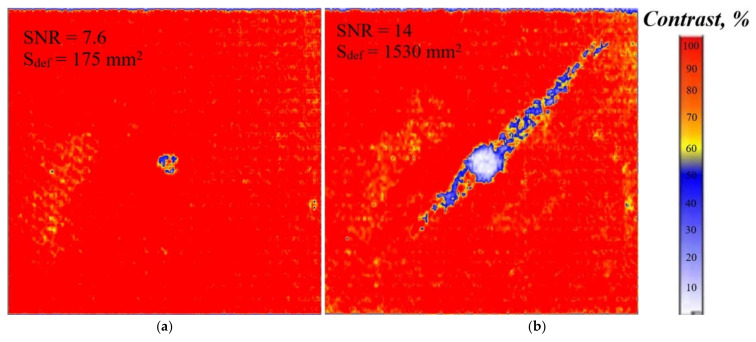
Ultrasonic C-scan results: (**a**)-delamination depth 0.1 mm (inspecting front surface), (**b**)-delamination depth 1.4 mm (inspecting rear surface).

**Figure 8 materials-15-07176-f008:**
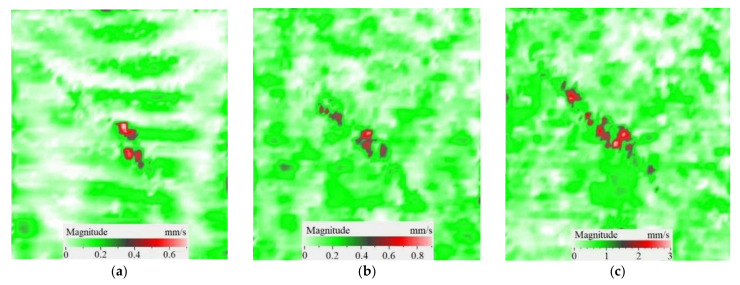
Total pattern of resonance vibrations on rear surface of CFRP sample with 18 J impact damage: (**a**)-7.44 kHz, (**b**)-32.8 kHz, (**c**)-46.5 kHz.

**Figure 9 materials-15-07176-f009:**
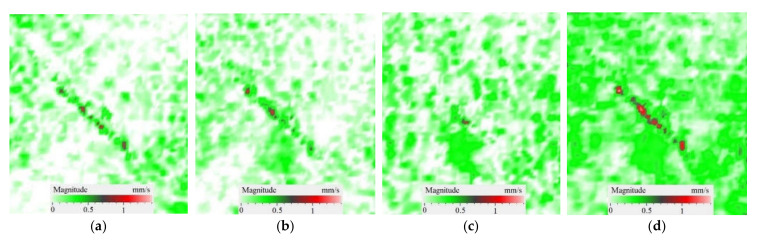
Resonance vibrations in CFRP composite with 18 J impact damage by *x* (**a**), *y* (**b**) and *z* coordinate (**c**) and total pattern at frequency of 43.6 kHz (**d**).

**Figure 10 materials-15-07176-f010:**
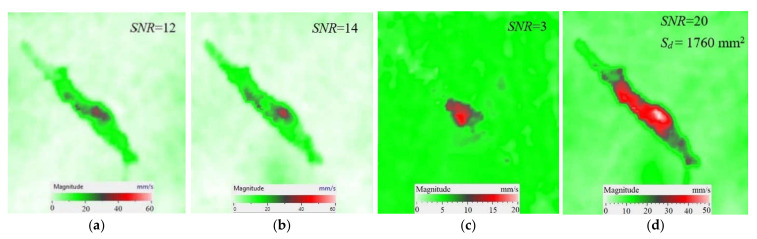
Frequency band data of CFRP sample by *x* (**a**), *y* (**b**), *z* (**c**) and *x, y, z* (**d**) coordinates.

**Table 1 materials-15-07176-t001:** Experimental and theoretical resonance harmonics for rectangular FBHs in the 50 Hz–13 kHz frequency range by *x*, *y*, *z* coordinates.

Frequency	*x*, *y*, *z-*Coordinates	*x*	*y*	*z*
SLDV 2.4 Hz–2.7 kHz Comsol 2.69 kHz	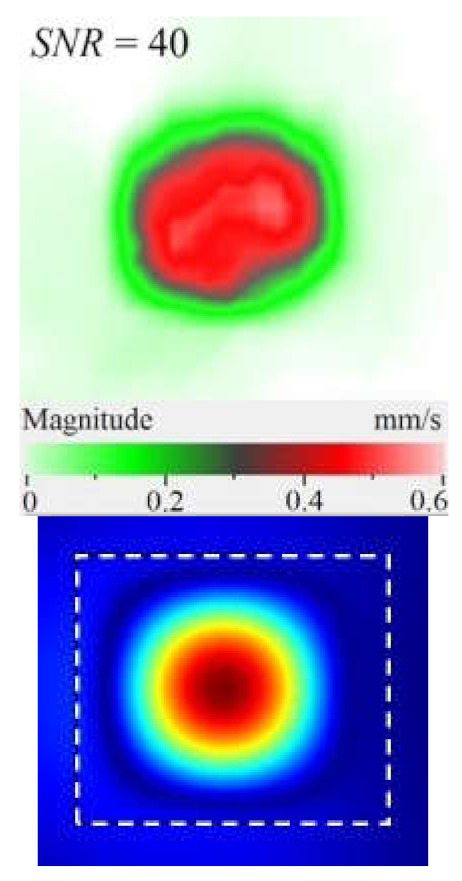	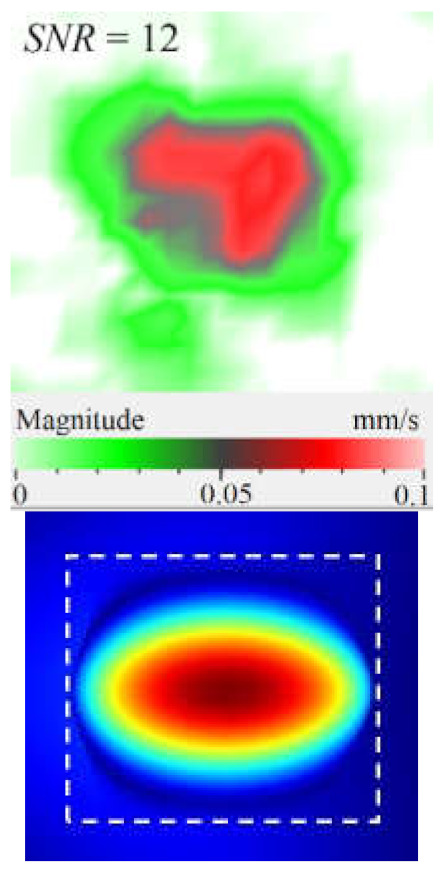	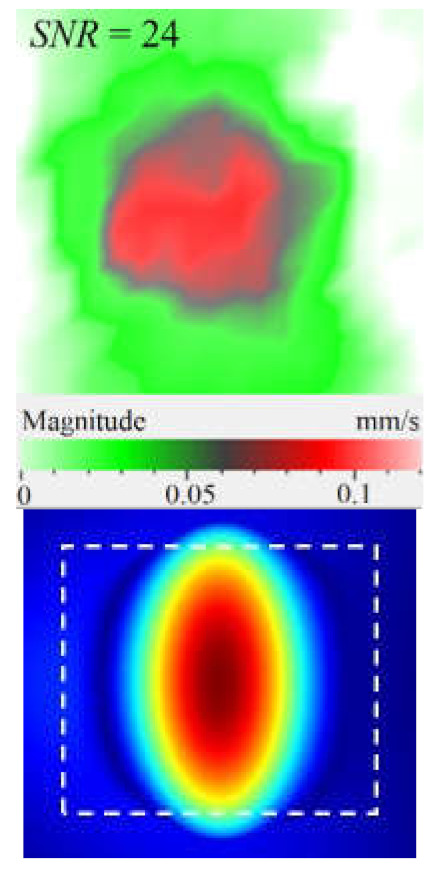	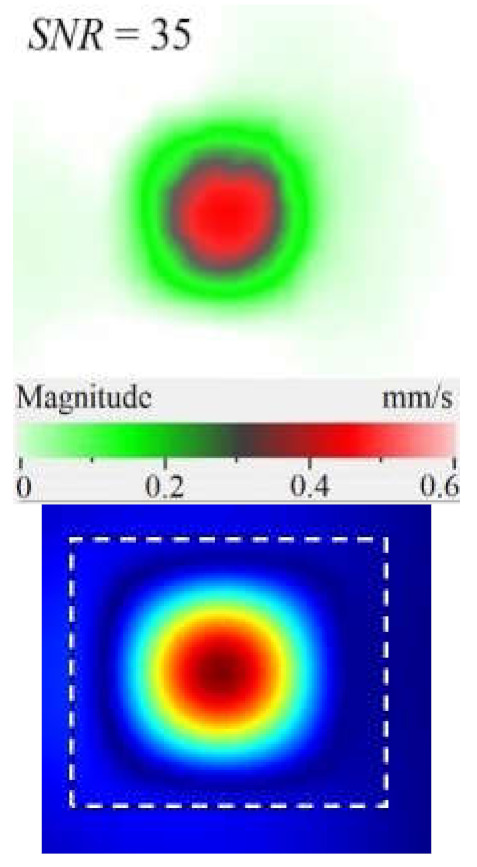
SLDV 3.9 kHz– 5.9 kHz Comsol 5.03 kHz	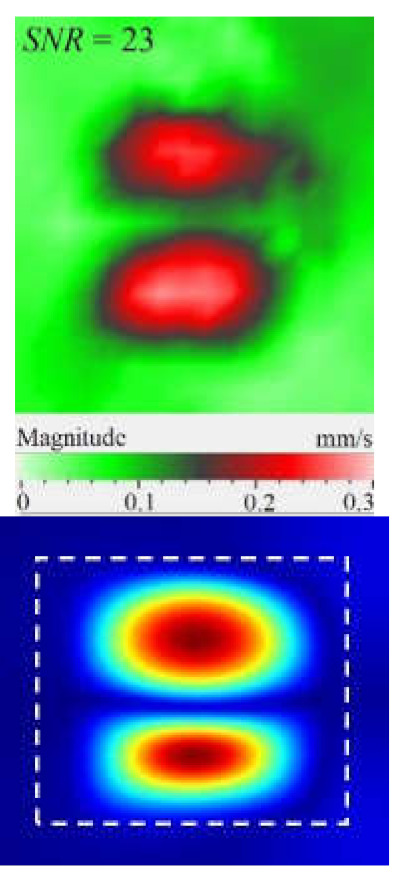	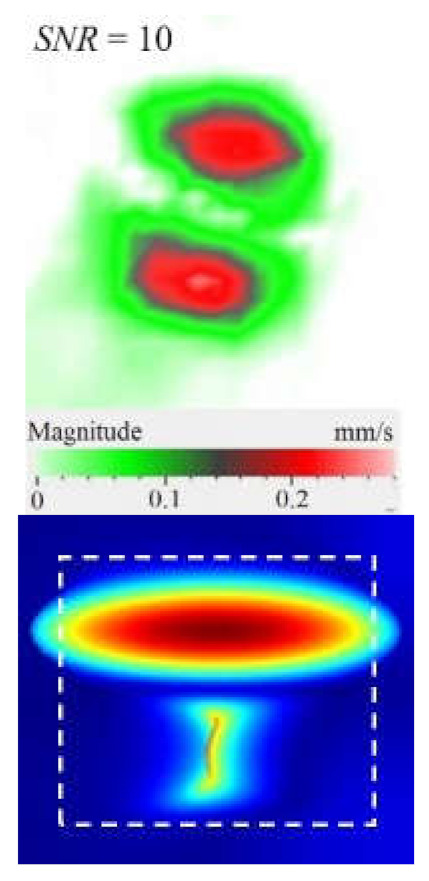	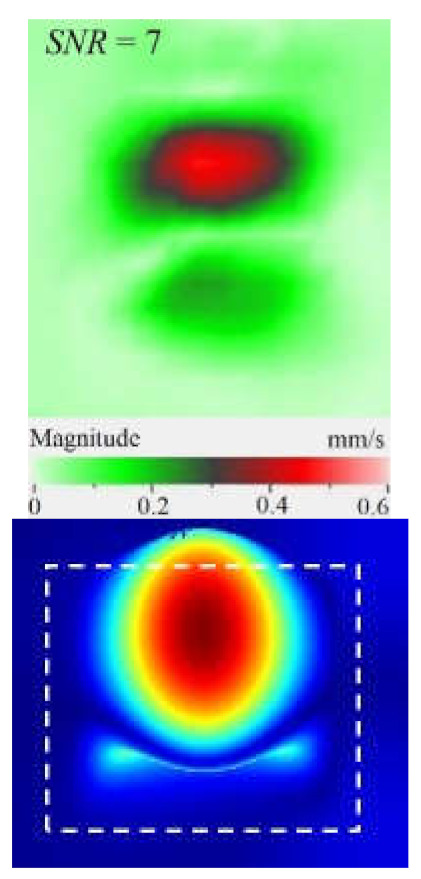	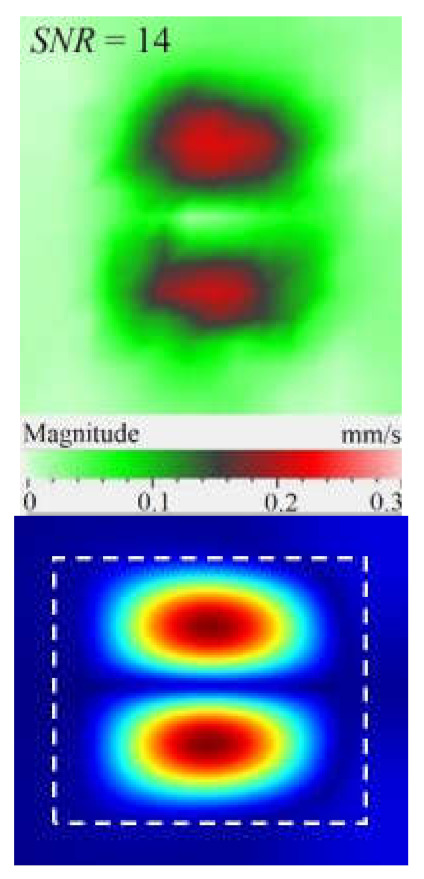
SLDV 9.2 kHz– 9.4 kHz Comsol 8.6 kHz	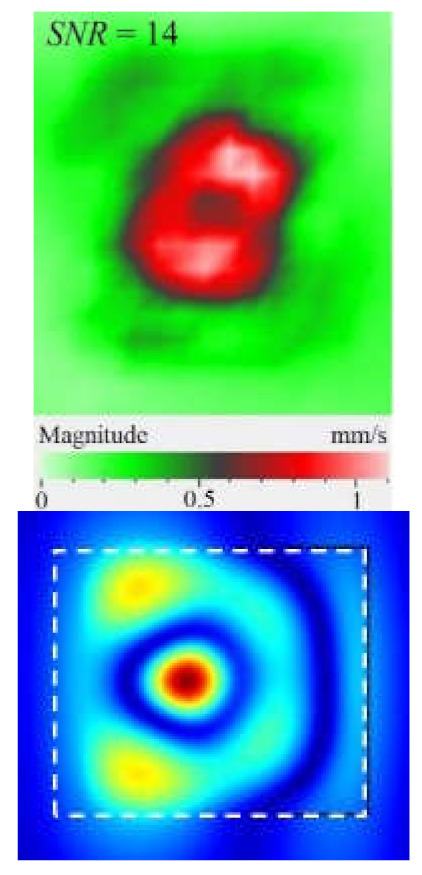	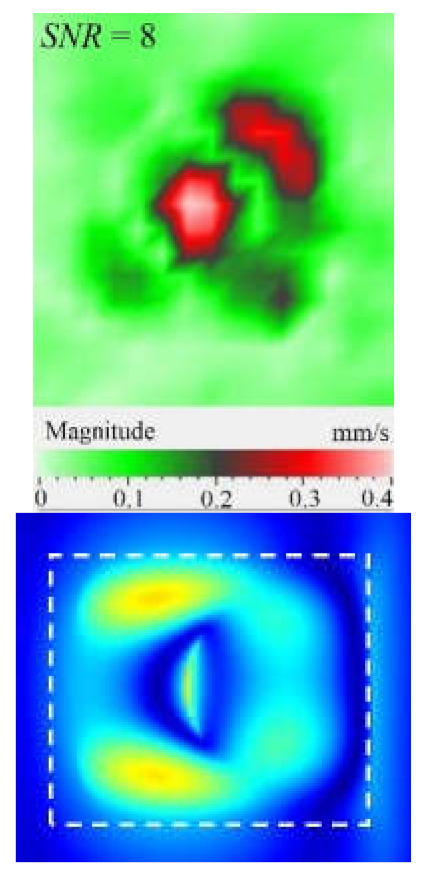	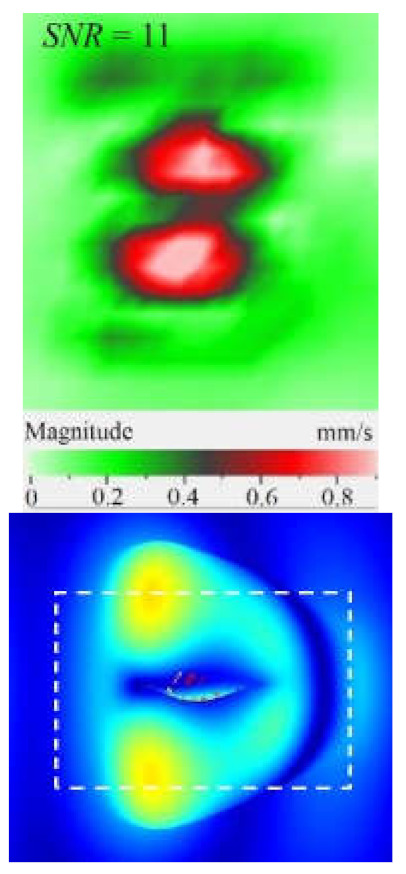	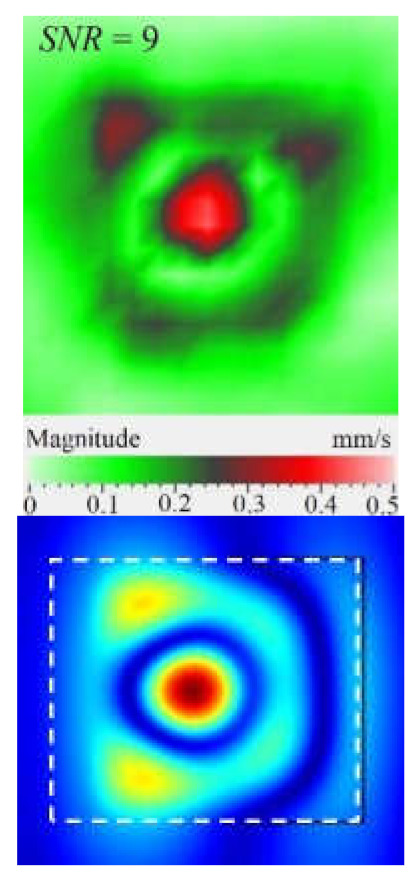
Frequency band data, sweep from 30 Hz to 100 kHz	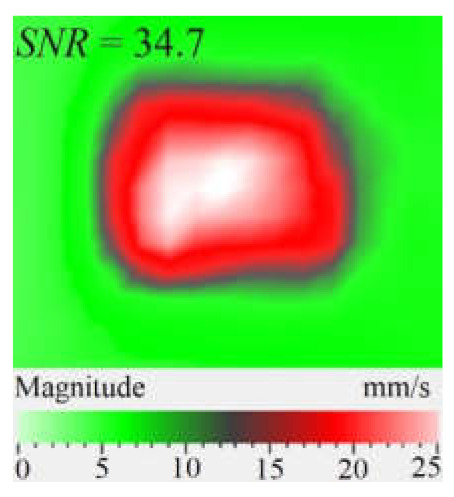	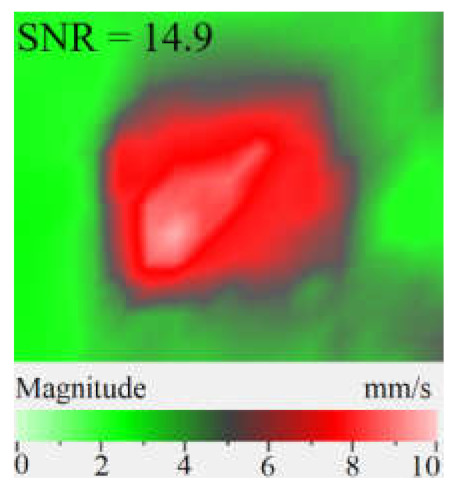	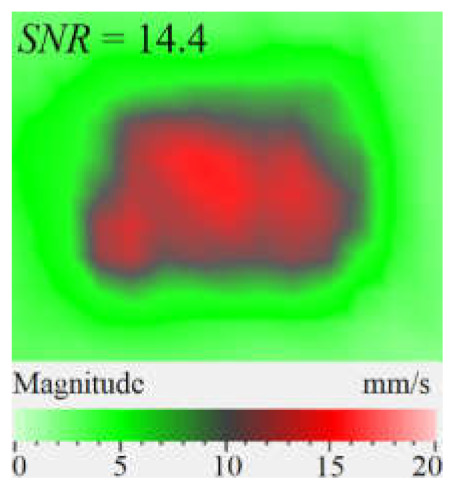	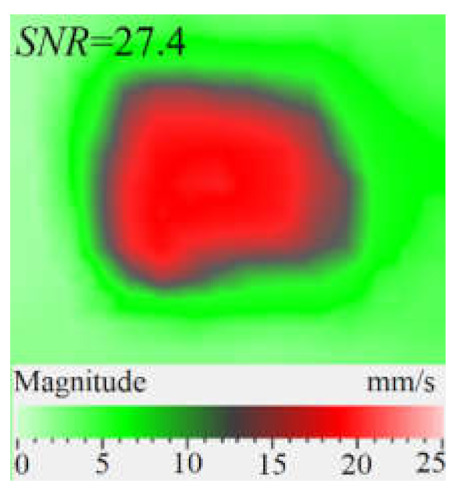

## Data Availability

Not applicable.

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
