# Peer review of "Detecting Defects in Composite Polymers by Using 3D Scanning Laser Doppler Vibrometry"

_materials, 2022, doi:10.3390/ma15207176_

Round 1

Reviewer 1 Report

This manuscript explores the potential of laser doppler vibrometry as a non-destructive measuring technique for analyzing defects in composite material components. The experimental results show a palpable contrast between vibration at the defect and non-defect areas at resonance frequencies. Two tests were performed: a PMMA with a flat bottom hole defect and a defect produced by an impact on carbon-fiber-reinforced polymer. The introduction is sound, but there are some issues along the paper to work on to enhance the resulting document. Please consider the following recommendations:

Section 2. Could the authors detail the measurement procedure? Did the authors repeat the tests? Regarding the measurement equipment, please include (if possible) the information about accuracy and repeatability.

Section 3. This section should be extended to inform about: Constitutive equation, type of elements, reasons for element sizes and their definition, boundary conditions, and computational time.

Section 4. This section is hard to follow because there are many repetitions, and it is difficult to identify some of the values cited in the text with the results portrayed in the figures.

Section 4. Please use the identical multiples for Amplification of wavefront vibrations units (some values appear in micrometers/s and others in mm/s).

Section 4. Table 1. SLDV and Comsol simulations do not have the same color legend. Furthermore, could the authors explain why the band from 6 kHz to 9 kHz is not included?

Section 4.1. Page 8, “A certain discrepancy between simulated and experimental results (up to 10% by frequency) was also observed at the higher frequencies.” How was this discrepancy evaluated?

Section 4.3. Did the authors consider including a simulation based on the 3D scanning?

Author Response

Please, find our reply to reviewer. Thank you

Reviewer 2 Report

In the paper, three components of vibration components are comparatively analyzed on defects with a simple geometry by modeling local resonance vibrations and performing 3D laser scanning. Basic investigations of resonance phenomena in objects with structural defects subjected to acoustic stimulation were conducted on polymeric materials, proving a nondestructive character of the LDR technique. It has been shown that, in the case of “classical” flat bottom hole-like defects, dominating are out-of-plane vibrations. Oppositely, the analysis of impact damage in carbon fiber reinforced polymer has revealed an essential contribution of in-plane vibration components. The article is well organized and logical. But the paper needs some refinement before it can be published. My detailed comments are as follows:

1.     Summary section is incomplete, missing part of the method, maybe you should improve it.

2.     The introduction of the article is a little simple. The introduction should be concise and comprehensive.

3.     LDR phenomenon appeared in the introduction of the article, but LDR phenomenon was not explained clearly.

4.     FBH tests: The magnitude of vibration amplification was evaluated by determining the signal-to-noise ratio (SNR), and what is the specific quantitative relationship between them? The illustration part lacks literature or experimental support.

5.      Fig. 4: The relation between the mean vibration velocity and coordinate is not clearly explained. The mean vibration velocity mentioned below is not directly reflected in the figure.

6.     The format of the figure 2 and the table 1 may be need to be standardized and adjusted.

7.     The author did not describe the future research plan in conclusion part, and where the future research of this article is directed. The conclusion is just a summary of the article, so no readers are excited about the research prospects.   

Author Response

(The authors gave the same response as above.)

Reviewer 3 Report

The authors have prepared a research article entitled “Detecting defects in composite polymers by using 3D scanning laser Doppler vibrometry”. The article has some interesting results and the authors have made considerable attention to preparing it. However, some issues need to be clarified before further consideration. Thus, the reviewer recommends this work can be published in Polymers after a major review.

1.       Rewrite the abstract. Remove very generic statements and focus directly on the original findings of the current article.

2.       English should be improved extensively throughout the manuscript.

3.       The introduction is very short and should be improved entirely so that the reader can identify the scientific advances of this work. The introduction should be emphasized some polymer composites that cause acoustic stimulation of defects in them. Focus on the mechanical properties of polymer composites. Thus, the current manuscript is missing some key references based on such composites. Therefore, the authors should cover the following references in appropriate sections.

https://doi.org/10.1021/acsbiomaterials.2c00786, https://doi.org/10.1016/j.compositesb.2022.110150

It would be more realistic to cover such kind of research work in the current manuscript. Which will enrich the quality of the current manuscript as well as inquisitiveness to the readers.

4.       Conclusions are very broad and they should be concise.

Author Response

(The authors gave the same response as above.)

Round 2

Reviewer 2 Report

The authors have made sufficient revisions according to the comments, and I suggest accepting the paper without further revisions.

Reviewer 3 Report

The authors have clarified all my concerns. The revised manuscript can be acceptable in its current form.